# Graph Normalizing Flows

**Jenny Liu**[*]
University of Toronto
Vector Institute
jyliu@cs.toronto.edu

**Aviral Kumar**[*†]
UC Berkeley
aviralk@berkeley.edu

**Jimmy Ba**
University of Toronto
Vector Institute
jba@cs.toronto.edu

**Jamie Kiros**
Google Research
kiros@google.com

**Kevin Swersky**
Google Research
kswersky@google.com

## Abstract

We introduce graph normalizing flows: a new, reversible graph neural network model for prediction and generation. On supervised tasks, graph normalizing flows perform similarly to message passing neural networks, but at a significantly reduced memory footprint, allowing them to scale to larger graphs. In the unsupervised case, we combine graph normalizing flows with a novel graph auto-encoder to create a generative model of graph structures. Our model is permutation-invariant, generating entire graphs with a single feed-forward pass, and achieves competitive results with the state-of-the art auto-regressive models, while being better suited to parallel computing architectures.

## 1 Introduction

Graph-structured data is ubiquitous in science and engineering, and modeling graphs is an important component of being able to make predictions and reason about these domains. Machine learning has recently turned its attention to modeling graph-structured data using graph neural networks (GNNs) [8, 23, 16, 13, 6] that can exploit the structure of relational systems to create more accurate and generalizable predictions. For example, these can be used to predict the properties of molecules in order to aid in search and discovery [5, 6], or to learn physical properties of robots such that new robots with different structures can be controlled without re-learning a control policy [27].

In this paper, we introduce a new formulation for graph neural networks by extending the framework of normalizing flows [22, 3, 4] to graph-structured data. We call these models graph normalizing flows (GNFs). GNFs have the property that the message passing computation is exactly reversible, meaning that one can exactly reconstruct the input node features from the GNN representation; this results in GNFs having several useful properties.

In the supervised case, we leverage a similar mechanism to [7] to obtain significant memory savings in a model we call reversible graph neural networks, or GRevNets. Ordinary GNNs require the storage of hidden states after every message passing step in order to facilitate backpropagation. This means one needs to store $O(\#\text{nodes} \times \#\text{message passing steps})$ states, which can be costly for large graphs. In contrast, GRevNets can reconstruct hidden states in lower layers from higher layers during backpropagation, meaning one only needs to store $O(\#\text{nodes})$ states. A recent approach for memory saving based on recurrent backpropagation (RBP) [1, 20, 18] requires running message passing to convergence, followed by the approximate, iterative inversion of a large matrix. Conversely, GRevNets get the exact gradients at a minor additional cost, equivalent to one extra forward pass. We

---

[*]Equal Contribution
[†]Work done during an internship at Google

show that GRevNets are competitive with conventional memory-inefficient GNNs, and outperform RBP on standard benchmarks.

In the unsupervised case, we use GNFs to develop a generative model of graphs. Learned generative models of graphs are a relatively new and less explored area. Machine learning has been quite successful at generative modeling of complex domains such as images, audio, and text. However, relational data poses new and interesting challenges such as permutation invariance, as permuting the nodes results in the same underlying graph.

One of the most successful approaches so far is to model the graph using an auto-regressive process [17, 30]. These generate each node in sequence, and for each newly generated node, the corresponding edges to previously generated nodes are also created. In theory, this is capable of modeling the full joint distribution, but computing the full likelihood requires marginalizing over all possible node-orderings. Sequential generation using RNNs also potentially suffers from trying to model long-range dependencies.

Normalizing flows are primarily designed for continuous-valued data, and the GNF models a distribution over a structured, continuous space over sets of variables. We combine this with a novel permutation-invariant graph auto-encoder to generate embeddings that are decoded into an adjacency matrix in a similar manner to [15, 19]. The result is a fully permutation-invariant model that achieves competitive results compared to GraphRNN [30], while being more well-suited to parallel computing architectures.

## 2 Background

### 2.1 Graph Neural Networks

**Notation:** A graph is defined as $\mathcal{G} = (H, \Omega)$, where $H \in \mathbb{R}^{N \times d_n}$, $H = (\mathbf{h}^{(1)}, \cdots, \mathbf{h}^{(N)})$ is the node feature matrix consisting of node features, of size $d_n$, for each of the $N$ nodes ($\mathbf{h}^{(v)}$ for node $v$) in the graph. $\Omega \in \mathbb{R}^{N \times N \times (d_e + 1)}$ is the edge feature matrix for the graph. The first channel of $\Omega$ is the *adjacency matrix* of the graph (i.e. $\Omega_{i,j,0} = 1$ if $e_{ij}$ is an edge in the graph). The rest of the matrix $\Omega_{i,j,1:(d_e+1)}$ is the set of edge features of size $d_e$ for each possible edge $(i, j)$ in the graph.

Graph Neural Networks (GNNs) or Message Passing Neural Nets (MPNNs) [6] are a generalization/unification of a number of neural net architectures on graphs used in literature for a variety of tasks ranging from molecular modeling to network relational modeling. In general, MPNNs have two phases in the forward pass – a message passing (MP) phase and a readout (R) phase. The MP phase runs for $T$ time steps, $t = 1, \ldots, T$ and is defined in terms of message generation functions $M_t$ and vertex update functions $U_t$. During each step in the message passing phase, hidden node features $\mathbf{h}_t^{(v)}$ at each node in the graph are updated based on messages $\mathbf{m}_{t+1}^{(v)}$ according to

$$\mathbf{m}_{t+1}^{(v)} = \mathsf{Agg}\left(\left\{M_t(\mathbf{h}_t^{(v)}, \mathbf{h}_t^{(u)}, \Omega_{u,v})\right\}_{u \in \mathcal{N}(v)}\right) \tag{1}$$

$$\mathbf{h}_{t+1}^{(v)} = U_t(\mathbf{h}_t^{(v)}, \mathbf{m}_{t+1}^{(v)}) \tag{2}$$

where Agg is an aggregation function (e.g., sum), and $\mathcal{N}(v)$ denotes the set of neighbours to node $v$ in the graph. The R phase converts the final node embeddings at MP step $T$ into task-specific features by e.g., max-pooling.

One particularly useful aggregation function is graph attention [26], which uses attention [2, 25] to weight the messages from adjacent nodes. This involves computing an attention coefficient $\alpha$ between adjacent nodes using a linear transformation $W$, an attention mechanism $a$, and a nonlinearity $\sigma$,

$$e_{t+1}^{(v,u)} = a(W\mathbf{h}_t^{(v)}, W\mathbf{h}_t^{(u)}), \quad \alpha_{t+1}^{v,u} = \frac{\exp(e_{t+1}^{(v,u)})}{\sum_{w \in \mathcal{N}(v)} \exp(e_{t+1}^{(u,w)})}$$

$$\mathbf{m}_{t+1}^{(v)} = \sigma\left(\sum_{u \in \mathcal{N}(v)} \alpha_{t+1}^{(v,u)} M(\mathbf{h}_t^{(v)}, \mathbf{h}_t^{(u)}, \Omega_{u,v})\right)$$

Multi-headed attention [25] applies attention with multiple weights $W$ and concatenates the results.

## 2.2 Normalizing Flows

Normalizing flows (NFs) [22, 3, 4] are a class of generative models that use invertible mappings to transform an observed vector $\mathbf{x} \in \mathbb{R}^d$ to a latent vector $\mathbf{z} \in \mathbb{R}^d$ using a mapping function $\mathbf{z} = f(\mathbf{x})$ with inverse $\mathbf{x} = f^{-1}(f(\mathbf{x}))$. The change of variables formula relates a density function over $\mathbf{x}$, $P(\mathbf{x})$ to one over $\mathbf{z}$ by

$$P(\mathbf{z}) = P(\mathbf{x}) \left| \frac{\partial f(\mathbf{x})}{\partial \mathbf{x}} \right|^{-1}$$

With a sufficiently expressive mapping, NFs can learn to map a complicated distribution into one that is well modeled as a Gaussian; the key is to find a mapping that is expressive, but with an efficiently computable determinant. We base our formulation on non-volume preserving flows, a.k.a RealNVP [4]. Specifically, the affine coupling layer involves partitioning the dimensions of $\mathbf{x}$ into two sets of variables, $\mathbf{x}^{(0)}$ and $\mathbf{x}^{(1)}$, and mapping them onto variables $\mathbf{z}^{(0)}$ and $\mathbf{z}^{(1)}$ by

$$\mathbf{z}^{(0)} = \mathbf{x}^{(0)}$$
$$\mathbf{z}^{(1)} = \mathbf{x}^{(1)} \odot \exp(s(\mathbf{x}^{(0)})) + t(\mathbf{x}^{(0)})$$

Where $s$ and $t$ are nonlinear functions and $\odot$ is the Hadamard product. The resulting Jacobian is lower triangular and its determinant is therefore efficiently computable.

# 3 Methods

## 3.1 Reversible Graph Neural Networks (GRevNets)

GRevNets are a family of reversible message passing neural network models. To achieve reversibility, the node feature matrix of a GNN is split into two parts along the feature dimension–$H_t^{(0)}$ and $H_t^{(1)}$. For a particular node in the graph $v$, the two parts of its features at time $t$ in the message passing phase are called $\mathbf{h}_t^0$ and $\mathbf{h}_t^1$ respectively, such that $\mathbf{h}_t^{(v)} = \text{concat}(\mathbf{h}_t^0, \mathbf{h}_t^1)$.

One step of the message passing procedure is broken down into into two intermediate steps, each of which is denoted as a half-step. $F_1(\cdot)$, $F_2(\cdot)$, $G_1(\cdot)$, and $G_2(\cdot)$ denote instances of the MP transformation given in Equations (1) and (2), with $F_1/G_1$ and $F_2/G_2$ indicating whether the function is applied to scaling or translation. These functions consist of applying $M_t$ and then $U_t$ to one set of the partitioned features, given the graph adjacency matrix $\Omega$. Figure 1 depicts the procedure in detail.

$$H_{t+\frac{1}{2}}^{(0)} = H_t^{(0)} \odot \exp\left(F_1\left(H_t^{(1)}\right)\right) + F_2(H_t^{(1)}) \qquad H_{t+1}^{(0)} = H_{t+\frac{1}{2}}^{(0)}$$
$$H_{t+\frac{1}{2}}^{(1)} = H_t^{(1)} \qquad\qquad H_{t+1}^{(1)} = H_{t+\frac{1}{2}}^{(1)} \odot \exp\left(G_1\left(H_{t+\frac{1}{2}}^{(0)}\right)\right) + G_2\left(H_{t+\frac{1}{2}}^{(0)}\right)$$

$$(3)$$

This architecture is easily reversible given $H_{t+1}^{(0)}$ and $H_{t+1}^{(1)}$, with the reverse procedure given by,

$$H_{t+\frac{1}{2}}^{(0)} = H_{t+1}^{(0)} \qquad\qquad\qquad H_t^{(1)} = H_{t+\frac{1}{2}}^{(1)}$$
$$H_{t+\frac{1}{2}}^{(1)} = \frac{\left(H_{t+1}^{(1)} - G_2\left(H_{t+\frac{1}{2}}^{(0)}\right)\right)}{\exp\left(G_1\left(H_{t+\frac{1}{2}}^{(0)}\right)\right)} \qquad\qquad H_t^{(0)} = \frac{\left(H_{t+\frac{1}{2}}^{(0)} - F_2(H_t^{(1)})\right)}{\exp\left(F_1\left(H_t^{(1)}\right)\right)} \qquad (4)$$

## 3.2 GNFs for Structured Density Estimation

In the same spirit as NFs, we can use the change of variables to give us the rule for exact density transformation. If we assume $H_t \sim P(H_t)$, then the density in terms of $P(H_{t-1})$ is given by $P(H_{t-1}) = \det \left| \frac{\partial H_{t-1}}{\partial H_t} \right| P(H_t)$ so that

$$P(\mathcal{G}) = \det \left| \frac{\partial H_T}{\partial H_0} \right| P(H_T) = P(H_T) \prod_{t=1}^{T} \det \left| \frac{\partial H_t}{\partial H_{t-1}} \right|$$

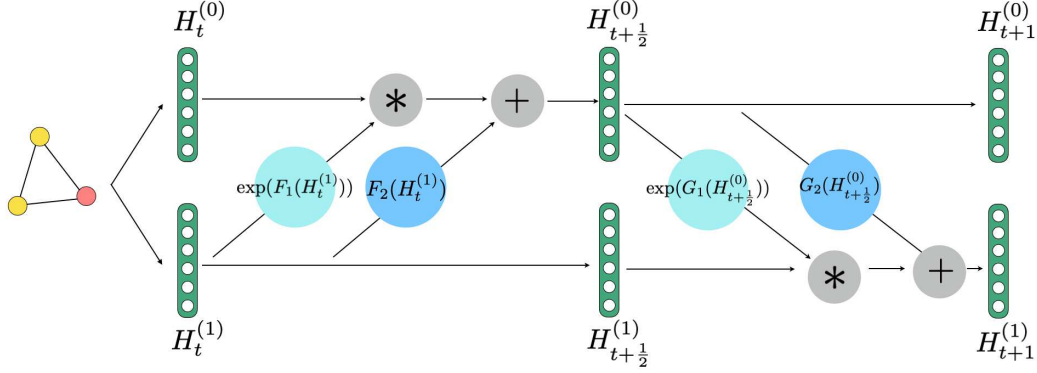

Figure 1: Architecture of 1 step of message passing in a GRevNet: $H_t^{(0)}$, $H_t^{(1)}$ denote the two parts of the node-features of a particular node. $F_1(\cdot), F_2(\cdot)$ and $G_1(\cdot), G_2(\cdot)$ are 1-step MP transforms consisting of applying $M_t$ and $U_t$ once each, with $F_1, G_1$ performing scaling and $F_2, G_2$ performing translation.

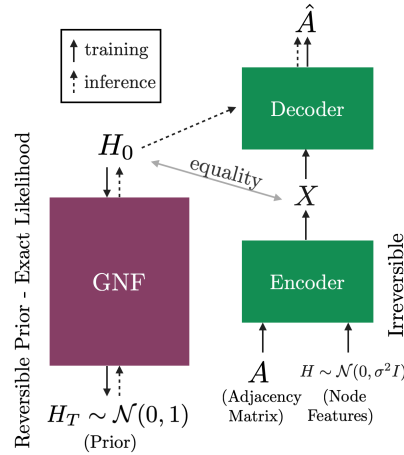

Figure 2: A summary of our graph generation pipeline using GNFs. The learned node features $X$ from the auto-encoder are used to train the GNF. At generation time, the GNF generates node features which are then fed into the decoder to get the predicted adjacency matrix.

with $H_0$ being the input node features. The Jacobians are given by lower triangular matrices, hence making density computations tractable.

GNFs can model expressive distributions in continuous spaces over *sets* of vectors. We choose the prior $P(H_T) = \prod_{i=1}^{N} \mathcal{N}(\mathbf{h}_i|0, I)$ to be a product of independent, standard Gaussian distributions. Sampling simply involves sampling a set of Gaussian vectors and running the inverse mapping. One free variable is the number of nodes that must be generated before initiating message passing. We simply model this as a fixed prior $P(N)$, where the distribution is given by the empirical distribution in the training set. Sampling graphs uniformly from the training set is equivalent to sampling $N$ from this distribution, and then sampling $\mathcal{G}$ uniformly from the set of training graphs with $N$ nodes.

Notice that the graph message passing induces dependencies between the nodes in the input space, which is reflected in the Jacobian. This also allows us to cast the RealNVP in the GNF framework: simply remove the edges from the graph so that all nodes become independent. Then each node transformation will be a sequence of reversible non-linear mappings, with no between-node dependencies[3]. We use this as a baseline to demonstrate that the GNF benefits when the nodes must model dependencies between each other.

In the absence of a known graph structure, as is the case for generation, we use a fully connected graph neural network. This allows the model to learn how to organize nodes in order to match a specific distribution. However, this poses a problem for certain aggregation functions like sum and mean. In the sum case, the message variance will increase with the number of nodes, and in both sum and mean cases, the messages from each node will have to contend with the messages from every other node. If there is a salient piece of information being sent from one node to another, then it could get drowned out by less informative messages. Instead, we opt to use graph attention as discussed in Section 2.1. This allows each node to choose the messages that it deems to be the most informative.

The result of using a fully connected graph is that the computational cost of message passing is $O(N^2)$, similar to the GraphRNN. However, each step of the GNF is expressible in terms of matrix operations, making it more amenable to parallel architectures. This is a similar justification for using transformers over RNNs [25].

## 3.3 Graph Auto-Encoders

While GNFs are expressive models for structured, continuous spaces, our objective is to train a generative model of graph structures, an inherently discrete problem. Our strategy to solve this is to use a two-step process: (1) train a permutation invariant graph auto-encoder to create a graph encoder that embeds graphs into a continuous space; (2) train a GNF to model the distribution of the graph embeddings, and use the decoder to generate graphs. Each stage is trained separately. A similar strategy has been employed in prior work on generative models in [15, 19].

Note that in contrast to the GraphVAE [12], which generates a single vector to model the entire graph, we instead embed a set of nodes in a graph jointly, but each node will be mapped to its own embedding vector. This avoids the issue of having to run a matching process in the decoder.

The graph auto-encoder takes in a graph $\mathcal{G}$ and reconstructs the elements of the adjacency matrix, $A$, where $A_{ij} = 1$ if node $v_i$ has an edge connecting it to node $v_j$, and 0 otherwise. We focus on undirected graphs, meaning that we only need to predict the upper (or lower) triangular portion of $A$, but this methodology could easily extend to directed graphs.

The encoder takes in a set of node features $H \in \mathbb{R}^{N \times d}$ and an adjacency matrix $A \in \{0,1\}^{N \times \frac{N}{2}}$ ($\frac{N}{2}$ since the graph is undirected) and outputs a set of node embeddings $X \in \mathbb{R}^{N \times k}$. The decoder takes these embeddings and outputs a set of edge probabilities $\hat{A} \in [0,1]^{N \times \frac{N}{2}}$. For parameters $\boldsymbol{\theta}$, we use the binary cross entropy loss function,

$$\mathcal{L}(\boldsymbol{\theta}) = -\sum_{i=1}^{N} \sum_{j=1}^{\frac{N}{2}} A_{ij} \log(\hat{A}_{ij}) + (1 - A_{ij}) \log(1 - \hat{A}_{ij}). \tag{5}$$

We use a relatively simple decoder. Given node embeddings $\mathbf{x}_i$ and $\mathbf{x}_j$, our decoder outputs the edge probability as

$$\hat{A}_{ij} = \frac{1}{1 + \exp(C(\|\mathbf{x}_i - \mathbf{x}_j\|_2^2 - 1))} \tag{6}$$

where $C$ is a temperature hyperparameter, set to 10 in our experiments. This reflects the idea that nodes that are close in the embedding space should have a high probability of being connected.

The encoder is a standard GNN with multi-head dot-product attention, that uses the adjacency matrix $A$ as the edge structure (and no additional edge features). In order to break symmetry, we need some way to distinguish the nodes from each other. If we are just interested in learning structure, then we do not have access to node features, only the adjacency matrix. In this case, we generate node features $H$ using random Gaussian variables $\mathbf{h}_i \sim \mathcal{N}(0, \sigma^2 I)$, where we use $\sigma^2 = 0.3$. This allows the graph network to learn how to appropriately separate and cluster nodes according to $A$. We generate a new set of random features each time we encode a graph. This way, the graph can only rely on the features to break symmetry, and must rely on the graph structure to generate a useful encoding.

Putting the GNF together with the graph encoder, we map training graphs from $H$ to $X$ and use this as training inputs for the GNF. Generating involves sampling $Z \sim \mathcal{N}(0, I)$ followed by inverting the GNF, $X = f^{-1}(Z)$, and finally decoding $X$ into $A$ and thresholding to get binary edges.

## 4 Supervised Experiments

In this section we study the capabilities of the supervised GNF, the GrevNet architecture.

**Datasets/Tasks:** We experiment on two types of tasks. Transductive learning tasks consist of semi-supervised document classification in citation networks (Cora and Pubmed datasets), where we test our model with the author's dataset splits [28], as well as 1% train split for a fair comparison against [18]. Inductive Learning tasks consist of PPI (Protein-Protein Interaction Dataset) [31] and QM9 Molecule property prediction dataset [21]. For transductive learning tasks we report classification accuracy, for PPI we report Micro F1 score, and for QM9, Mean Absolute Error (MAE). More details on datasets are provided in the supplementary material.

**Baselines:** We compare GRevNets to: (1) A vanilla GNN architecture with an identical architecture and the same number of message-passing steps; (2) Neumann-RBP [18] – which, to the best of our knowledge, is the state-of-the-art in the domain of memory-efficient GNNs.

## 4.1 Performance on benchmark tasks

Table 1 compares GRevNets to GNNs and Neumann RBP (NRBP). 1% train uses 1% of the data for training to replicate the settings in [18]. For these, we provide average numbers for GNN and GRevNet and **best** numbers for NRBP. The GRevNet architecture is competitive with a standard GNN, and outperforms NRBP.

We also compare GRevNets to the GAT architecture [26]. While GAT outperforms the naive GRevNet, we find that converting GAT into a reversible model by using it as the $F$ and $G$ functions within a GRevNet (GAT-GRevNet) only leads to a small drop in performance while allowing the benefits of a reversible model.

| | Dataset/Task | GNN | GRevNet | Neumann RBP | GAT | GAT-GRevNet |
|---|---|---|---|---|---|---|
| Cora | Semi-Supervised | 71.9 | **74.5** | 56.5 | **83.0** | 82.7 |
| Cora | (1% Train) | 55.5 | **55.8** | 54.6 | – | – |
| Pubmed | Semi-Supervised | **76.3** | 76.0 | 62.4 | **79.0** | 78.6 |
| Pubmed | (1% Train) | 76.6 | **77.0** | 58.5 | – | – |
| PPI | Inductive | **0.78** | 0.76 | 0.70 | – | – |

| Model | mu | alpha | HOMO | LUMO | gap | R2 |
|---|---|---|---|---|---|---|
| GNN | 0.474 | 0.421 | **0.097** | 0.124 | 0.170 | 27.150 |
| GrevNet | **0.462** | **0.414** | 0.098 | 0.124 | **0.169** | **26.380** |

| Model | ZPVE | U0 | U | H | G | Cv |
|---|---|---|---|---|---|---|
| GNN | 0.035 | 0.410 | **0.396** | **0.381** | 0.373 | 0.198 |
| GrevNet | 0.036 | **0.390** | 0.407 | 0.418 | **0.359** | **0.195** |

Table 1: Top: performance in terms of accuracy (Cora, Pubmed) and Micro F1 scores (PPI). For GNN and GrevNet, number of MP steps is fixed to 4. For Neumann RBP, we use 100 steps of MP. For GAT and GAT-GRevNet, we use 8 steps of MP. These values are averaged out over 3-5 runs with different seeds. Bottom: performance in terms of Mean Absolute Error (lower is better) for independent regression tasks on QM9 dataset. Number of MP steps is fixed to 4. The model was trained for 350k steps, as in [6].

## 4.2 Analysis of Memory Footprint

We first provide a more rigorous theoretical derivation for the memory footprint and then provide some quantitative results. Let us assume that the node feature dimension is $d$, and the maximum number of nodes in a graph is $N$. Let us assume weights (parameters) of the message passing function is a matrix of size $W$. For simplicity, assume a parameter-free aggregation function that sums over messages from neighbouring nodes. Finally, assume that the final classifier weights are $C$ in size. Suppose we run $K$ message passing steps. Total memory that needs to be allocated for a run of GNN (ignoring gradients for now; gradients will scale by a similar factor) is $W + C + K \times N \times d$ (= memory allotted to weights + intermediate graph-sized tensors generated + adjacency matrix). For a GNF, the total memory is $W + C + N \times d$. Note the *lack* of multiplicative dependence on the number of message passing steps in the latter term.

| MODEL | MOG (NLL) | MOG RING (NLL) | 6-HALF MOONS (NLL) |
|---|---|---|---|
| REALNVP | 4.2 | 5.2 | -1.2 |
| GNF | **3.6** | **4.2** | **-1.7** |

Table 2: Per-node negative log likelihoods (NLL) on synthetic datasets for REALNVP and GNF.

As a quantitative example, consider a semi-supervised classification task on the Pubmed network ($N = 19717, d = 500$). We assume that the message passing function for a GNN is as follows: FC(500) $\rightarrow$ ReLU() $\rightarrow$ FC(750) $\rightarrow$ ReLU() $\rightarrow$ FC(500). Each of the functions $F_1(\cdot)$ and $F_2(\cdot)$ (please see Figure 1 in the paper for notation) in the corresponding GNF have the following architecture: FC(250) $\rightarrow$ FC(750) $\rightarrow$ FC(250). We can compute the total memory allocated to weights/parameters: $W_{GNN} = 500 \times 750 + 750 \times 500, W_{GNF} = 2 \times (250 \times 750 + 750 \times 250)$. We perform $K = 5$ message passing steps for Pubmed. So, the amount of memory allocated to intermediate tensors in a GNN is $19717 \times 500 \times 5 + 19717 \times 750 \times 5$, and correspondingly for a GNF is $19717 \times 500$. Summing up, the overall memory requirements are: GNN = **945.9 M** and GNF = **80.2 M**. Hence, in this case, GNFs are at least $\geq 10\times$ memory efficient than GNNs. Further, we use self-attention in our experiments, which scales according to $\mathcal{O}(N^2)$. GNNs will store attention affinity matrices for each message passing step. In this case, a similar argument can show that this causes a difference of **11G** memory. When using 12G GPU machines, this difference is significant.

# 5 Unsupervised Experiments

## 5.1 Structured Density Estimation

We compare the performance of GNFs with RealNVP for structured density estimation on 3 datasets. Details of the model architecture can be found in the supplementary material.

**Datasets.** The first dataset is MIXTURE OF GAUSSIANS (MOG), where each training example is a set of 4 points in a square configuration. Each point is drawn from a separate isotropic Gaussian, so no two points should land in the same area. MIXTURE OF GAUSSIANS RING (MOG RING) takes each example from MOG and rotates it randomly about the origin, creating an aggregate training distribution that forms a ring. 6-HALF MOONS interpolates the original half moons dataset using 6 points with added noise.

**Results.** Our results are shown in Table 2. We outperform REALNVP on all three datasets. We also compare the generated samples of the two models on the MOG dataset in Figure 3.

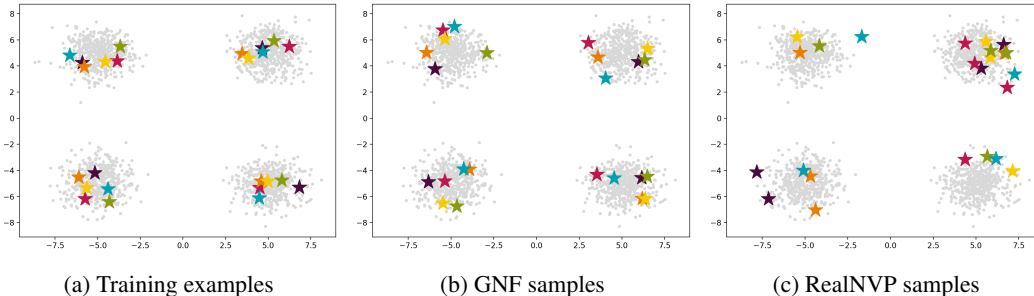

(a) Training examples          (b) GNF samples          (c) RealNVP samples

Figure 3: (a) shows the aggregate training distribution for the MOG dataset in gray, as well as 5 individual training examples. Each training example is shown in a different color and is a structured set of nodes where each node is drawn from a different Gaussian. (b) and (c) each show 5 generated samples from GNF and RealNVP, selected randomly. Each sample is shown in a different color. Note that, GNF learns to generate structured samples where each node resembles a sample from a different Gaussian, while RealNVP by design cannot model these dependencies. Best viewed in color.

## 5.2 Graph Generation

**Baselines.** We compare our graph generation model on two datasets, COMMUNITY-SMALL and EGO-SMALL from GraphRNN [30]. COMMUNITY-SMALL is a procedurally-generated set of 100 2-community graphs, where $12 \leq |V| \leq 20$. EGO-SMALL is a set of 200 graphs, where $4 \leq |V| \leq 18$,

|  | BINARY CE | | TOTAL # INCORRECT EDGES | | TOTAL # EDGES | |
|---|---|---|---|---|---|---|
| DATASET | TRAIN | TEST | TRAIN | TEST | TRAIN | TEST |
| EGO-SMALL | 9.8E-4 | 11E-04 | 24 | 32 | 3758 | 984 |
| COMMUNITY-SMALL | 5E-4 | 7E-04 | 10 | 2 | 1329 | 353 |

Table 3: Train and test binary cross-entropy (CE) as described in equation 5, averaged over the total number of nodes. TOTAL # INCORRECT EDGES measures the number of incorrect edge predictions (either missing or extraneous) in the reconstructed graphs over the entire dataset. TOTAL # EDGES lists the total number of edges in each dataset. As we use Gaussian noise for initial node features, we averaged 5 runs of our model to obtain these metrics.

drawn from the larger Citeseer network dataset [24]. For all experiments described in this section, we used scripts from the GraphRNN codebase [29] to generate and split the data. 80% of the data was used for training and the remainder for testing.

### 5.2.1 Graph Auto-Encoder

We first train a graph auto-encoder with attention, as described in Section 3.3. Every training epoch, we generate new Gaussian noise features for each graph as input to the encoder. The GNN consists of 10 MP steps, where each MP step uses a self-attention module followed by a multi-layer perceptron. Additional details can be found in the supplementary material.

Table 3 shows that our auto-encoder generalizes well to unseen test graphs, with a small gap between train and test cross-entropy. The total # of incorrect edges metric shows that the model achieves good test reconstruction on EGO-SMALL and near-perfect test reconstruction on COMMUNITY-SMALL.

### 5.2.2 Graph Normalizing Flows for Permutation Invariant Graph Generation

Our trained auto-encoder gives us a distribution over node embeddings that are useful for graph reconstruction. We then train a GNF to maximize the likelihood of these embeddings using an isotropic Gaussian as the prior. Once trained, at generation time the model flows $N$ random Gaussian embeddings sampled from the prior to $N$ node embeddings that describe a graph adjacency when run through the decoder.

Our GNF consists of 10 MP steps with attention and an MLP for each of $F_1$, $F_2$, $G_1$, and $G_2$. For more details on the architecture see the supplementary material.

**Evaluating Generated Graphs.** We evaluate our model by providing visual samples and by using the quantitative evaluation technique in GraphRNN [30], which calculates the MMD distance [9] between the generated graphs and a previously unseen test set on three statistics based on degrees, clustering coefficients, and orbit counts. We use the implementation of GraphRNN provided by the authors to train their model and their provided evaluation script to generate all quantitative results.

In [29], the MMD evaluation was performed by using a test set of $N$ ground truth graphs, computing their distribution over $|V|$, and then searching for a set of $N$ generated graphs from a much larger set of samples from the model that closely matches this distribution over $|V|$. These results tend to exhibit considerable variance as the graph test sets were quite small.

To achieve more certain trends, we also performed an evaluation by generating 1024 graphs for each model and computing the MMD distance between this generated set of graphs and the ground truth test set. We report both evaluation settings in Table 4. We also report results directly from [30] on two other graph generation models, GRAPHVAE and DEEPGMG, evaluated on the same graph datasets.

**Results.** We provide a visualization of generated graphs from GRAPHRNN and GNF in Figure 4. As shown in Table 4, GNF outperforms GRAPHVAE and DEEPGMG, and is competitive with GRAPHRNN. Error margins for GNF and GRAPHRNN and a larger set of visualizations are provided in the supplementary material.

## 6 Conclusion

We propose GNFs, normalizing flows using GNNs based on the RealNVP, by making the message passing steps reversible. In the supervised case, reversibility allows for backpropagation without

|  | COMMUNITY-SMALL | | | EGO-SMALL | | |
| --- | --- | --- | --- | --- | --- | --- |
| MODEL | DEGREE | CLUSTER | ORBIT | DEGREE | CLUSTER | ORBIT |
| GRAPHVAE | 0.35 | 0.98 | 0.54 | 0.13 | 0.17 | 0.05 |
| DEEPGMG | 0.22 | 0.95 | 0.4 | 0.04 | 0.10 | 0.02 |
| GRAPHRNN | 0.08 | 0.12 | 0.04 | 0.09 | 0.22 | 0.003 |
| GNF | 0.20 | 0.20 | 0.11 | 0.03 | 0.10 | 0.001 |
| GRAPHRNN(1024) | **0.03** | **0.01** | **0.01** | 0.04 | 0.05 | 0.06 |
| GNF(1024) | 0.12 | 0.15 | 0.02 | **0.01** | **0.03** | **0.0008** |

Table 4: Graph generation results depicting MMD for various graph statistics between the test set and generated graphs. GRAPHVAE and DEEPGMG are reported directly from [30]. The second set of results (GRAPHRNN, GNF) are from evaluating the GraphRNN evaluation scheme with node distribution matching turned on. We trained 5 separate models of each type and performed 3 trials per model, then averaged the result over 15 runs. The third set of results (GRAPHRNN (1024), GNF (1024)) are obtained when evaluating on the test set over all 1024 generated graphs (no sub-sampling of the generated graphs based on node similarity). In this case, we trained and evaluated the result over 5 separate runs per model.

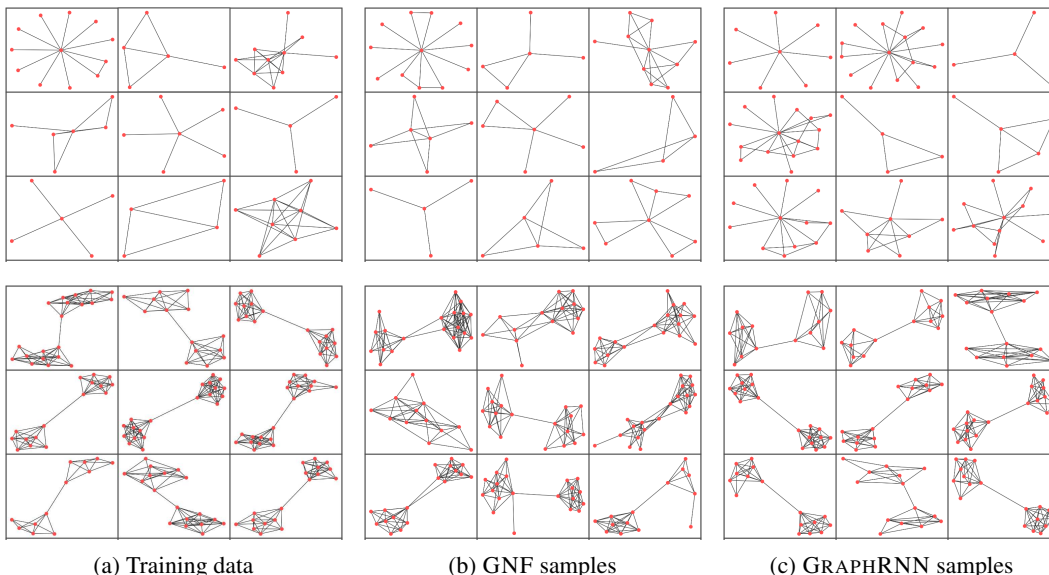

(a) Training data    (b) GNF samples    (c) GRAPHRNN samples

Figure 4: Dataset examples and samples, drawn randomly, from the generative models. Top row: EGO-SMALL, bottom row: COMMUNITY-SMALL.

the need to store hidden node states. This provides significant memory savings, further pushing the scalability limits of GNN architectures. On several benchmark tasks, GNFs match the performance of GNNs, and outperform Neumann RBP. In the unsupervised case, GNFs provide a flexible distribution over a set of continuous vectors. Using the pre-trained embeddings of a novel graph auto-encoder, we use GNFs to learn a distribution over the embedding space, and then use the decoder to generate graphs. This model is permutation invariant, yet competitive with the state-of-the-art auto-regressive GraphRNN model. Future work will focus on applying GNFs to larger graphs, and training the GNF and auto-encoder in an end-to-end approach.

# 7    Acknowledgements

We would like to thank Harris Chan, William Chan, Steve Kearnes, Renjie Liao, and Mohammad Norouzi for their helpful discussions and feedback, as well as Justin Gilmer for his help with the MPNN code.

## Footnotes

[3]This formulation is specifically applicable to unstructured vector spaces, as opposed to images, which would involve checkerboard partitions and other domain-specific heuristics.

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
