[Supplementary Material]

# Supplementary Material for Graph Normalizing Flows

## 1 Supplementary Material

### 1.1 Supervised Experiments Details

#### 1.1.1 Datasets Description

The following datasets were used in the experiments we reported in the main body of the paper.

- *Molecule property prediction on QM9* [5]: which consists of about 134k drug-like molecules made up of Hydrogen (H), Carbon (C), Oxygen (O), Nitrogen (N), and Flourine (F) atoms containing up to 9 heavy (non Hydrogen) atoms.

- *Semi-supervised document classification on citation networks*: A node of a network represents a document associated with a bag-of-words feature. Nodes are connected based on the citation links. Given a portion of nodes labeled with subject categories, e.g., science, history, the task is to predict the categories for unlabeled nodes within the same network. We use two citation networks from [7] - Cora and Pubmed. We try this with two settings - one with the author provided dataset splits into train/test/validation and the other with 1%/49%/50% train/test/validation splits.

- *Inductive Learning on Protein-Protein Interaction (PPI) Dataset*: PPI consists of graphs corresponding to different human tissues [9]. The dataset contains 20 graphs for training, 2 for validation and 2 for testing. Testing graphs remain completely unobserved during training. To construct the graphs, we used the preprocessed data provided by [2] and [6].

#### 1.1.2 Hyperparameter Tuning and Other Details

The following list describes major hyperparameter settings and some other implementation details for our model.

- *L2 parameter regularization:* In all models, for all runs, we applied a L2-regularization on the weights with a coefficient of 0.001.

- *Number of message passing steps:* We performed a search for the number of message passing steps over the following set - [1, 2, 4, 5, 10, 20] on Cora and Pubmed. We found that 4 works the best for GNN and GRevNet, and stuck to that for experiments on all datasets. For Neumann RBP, we tried 100 and 200 message passing steps, of which 100 worked better.

- *Selection of the test model:* We selected the test model by storing the model with the best performance in terms of accuracy/Micro F1 score/Mean *Squared* error (for QM9) on a held-out validation dataset.

- *Batch Normalization:* It is observed that as the number of message passing steps increases beyond a limit (in our case it was 20), the GNN/GRevNet model starts to perform worse,

| | COMMUNITY-SMALL | | | EGO-SMALL | | |
|---|---|---|---|---|---|---|
| **MODEL** | DEGREE | CLUSTER | ORBIT | DEGREE | CLUSTER | ORBIT |
| GRAPHVAE | 0.35 | 0.98 | 0.54 | 0.13 | 0.17 | 0.05 |
| DEEPGMG | 0.22 | 0.95 | 0.4 | 0.04 | 0.10 | 0.02 |
| GRAPHRNN | $0.08 \pm 0.06$ | $0.12 \pm 0.07$ | $0.04 \pm 0.04$ | $0.09 \pm 0.10$ | $0.22 \pm 0.16$ | $0.003 \pm 0.004$ |
| GNF | $0.20 \pm 0.07$ | $0.20 \pm 0.07$ | $0.11 \pm 0.07$ | $0.03 \pm 0.03$ | $0.10 \pm 0.05$ | $0.001 \pm 0.0009$ |
| GRAPHRNN(1024) | $\mathbf{0.03 \pm 0.02}$ | $\mathbf{0.01 \pm 0.0007}$ | $\mathbf{0.01 \pm 0.009}$ | $0.04 \pm 0.02$ | $0.05 \pm 0.02$ | $0.06 \pm 0.05$ |
| GNF(1024) | $0.12 \pm 0.006$ | $0.15 \pm 0.004$ | $0.02 \pm 0.003$ | $\mathbf{0.01 \pm 0.003}$ | $\mathbf{0.03 \pm 0.004}$ | $\mathbf{0.0008 \pm 0.0002}$ |

Table 1: Graph generation results showing MMD for various graph statistics between the test set and generated graphs. GRAPHVAE and DEEPGMG are reproduced directly from the GraphRNN paper. The second set of results (GRAPHRNN, GNF) are from running the GraphRNN evaluation script with node distribution matching turned on. We trained 5 separate models of each type and did 3 runs per models, then took the average over the 15 runs. The third set of results (GRAPHRNN (1024), GNF (1024)) are from evaluating on the test set and 1024 generated graphs. Again we trained 5 separate models of each type and evaluated the MMD over 5 separate runs, 1 run per model.

and often it is hard to optimize the whole system well – in an end-to-end manner. Likely, the whole model ends up at a bad optimum and is unable to recover from it. Similar observations were made by [4]. In order to tackle this problem, we applied batch norm at each of the layers during message passing. This helps with training up to about 40 steps. In the results with, 4 and 10 steps of message passing, we don't use Batch Normalization.

- *Optimization:* We used Adam Optimizer [3] for optimizing GNNs and GRevNets. We chose a fixed learning rate of 1e-4. Changing the learning rate to 1e-3, sometimes doesn't work and training is unstable. We applied gradient clipping, allowing a maximum gradient norm of 4.0 in all cases. For QM9, we chose a learning rate of 1e-3, as the authors specify in the MPNN paper [1]. For Neumann RBP, we found that Adam doesn't work well. So, we chose the settings specified by the author, that is SGD with Momentum of 0.9 and a learning rate of 1e-3.

- *Architecture Design:* For the message generation step of the message passing phase, we use an MLP over the node features. For the update step during message passing, we use a GRU-like update to update the node features. The final classifier/regressor on top of the graph net module was an MLP with 2 layers.

## 1.2 Unsupervised Experiment Details

### 1.2.1 Results with Error Bars

In Table 1, we show the results with error bars for GraphRNN and GNF. GraphVAE and DeepGMG are reported directly from [8].

### 1.2.2 More Graph Samples

In Figures 1 and 2 we show the full set of samples on the EGO-SMALL and COMMUNITY-SMALL datasets.

## 1.3 Computing Infrastructure

For all experiments in this section, we trained on a single GPU, either a Tesla P100 or Titan Xp.

### 1.3.1 Structured Density Estimation

We train a GNF with 12 message passing steps. We apply batch norm to the input at the beginning of each step, and then we use the same module for $F_1$, $F_2$, $G_1$, and $G_2$. The module consists of a dot-product multi-head self-attention layer followed by an MLP with 5 layers, latent dimension of 256, and ReLu non-linearities. We use 8 attention heads.

For RealNVP, we use an analogous architecture, with 12 coupling layers, batch norm at the beginning of each step followed by an MLP of 5 layers with latent dimension 256 and ReLu non-linearities.

(a) Training data          (b) GNF samples          (c) GRAPHRNN samples

Figure 1: Left, training data graphs from EGO-SMALL. Middle, generated graphs from GNF. Right, generated graphs from GRAPHRNN. Samples were picked at random.

(a) Training data          (b) GNF samples          (c) GRAPHRNN samples

Figure 2: Left, training data graphs from COMMUNITY-SMALL. Middle, generated graphs from GNF. Right, generated graphs from GRAPHRNN. Samples were picked at random.

We train both models for 15k steps using the Adam optimizer with a learning rate of 1e-04.

### 1.3.2  Graph Autoencoder

We found that EGO-SMALL and COMMUNITY-SMALL needed differing capacities for the node embedding. We used an embedding size of 14 for EGO-SMALL and 30 for COMMUNITY-SMALL. We used 10 message passing steps. Each step uses the same architecture, a batch norm layer, followed by multi-head dot-product self-attention, and then an MLP with 3 layers, a latent dimension of 2048,

and ReLu non-linearities. We used 8 attention heads. We shared weights between message passing steps. For both datasets we trained for 100k steps using the Adam Optimizer and a learning rate of 1e-04. We use an exponential learning rate decay of 0.99 every 1000 steps.

### 1.3.3 GNF for Graph Generation

We use the same embedding sizes as the graph autoencoder, 14 for EGO-SMALL and 30 for COMMUNITY-SMALL. We used 12 message passing steps. For each message passing step we used the same architecture for each $F_1$, $F_2$, $G_1$ and $G_2$. We have a batch norm layer followed by a multi-head dot-product self-attention module, then an MLP with 3 layers, a latent dimension of 2048, and ReLu non-linearities. We used 8 attention heads. We did not share weights between message passing steps. For both datasets we train for 100k steps using the Adam Optimizer and a learning rate of 1e-04 for EGO-SMALL and 1e-05 for COMMUNITY-SMALL. We use an exponential learning rate decay of 0.99 every 1000 steps.