[Reviews · NeurIPS 2019]

Reviewer 1



Originality: The task is not new. But the paper did combined the normalizing flow to graph neural network in a new way. Related works are cited. Quality: The paper showed many results to support their model. They compared their method with several existing method and showed that their method is either better or as good as other methods. However, the results for the generative task seems to suggest that their method is not comparable to graphRNN on at least one task(community-small) and they didn't try to analysis that. Also they claimed that their method has better memory usage but didn't provide any quantitive results to show that. Clarity: The paper is well organized. Related works are introduced in detail. Significance: Good but results are on small datasets. To have better impact and to really support their memory advantage claim they need results for large graph.

Reviewer 2



Significance: 1. The paper is well written and easy to follow. All experimental details are provided along with the code. 2. As per my knowledge, this is the first work which replaces non-linear functions in reversible normalizing flow model with GNN model. This simple change led to development of reversible model for graph processing. 3. On the generation side - Although the computational cost is same as GraphRNN O(N^2), inference model for graph generation can be parallelized. 4. Drawback - Graph generation is shown on very small dataset with max of 20 nodes. It would be better to compare against standard baselines as reported in GraphRNN work such as Grid, Protein and medium sized community / ego graphs. 5. Moreover, the results in Table 1 & 4 is not overwhelming. Clarification: 1. On optimizing using Eq.(5), there is a good possibility that the loss function is dominated by large chunk of non-edge terms. Have you considered balancing the loss for positive and negative edges ?

Reviewer 3



The paper introduced a graph formulation of normalizing flows. The model is clearly presented and justified. The numerical experiments are less convincing than the presentation of the model. The results are marginally better than the baselines. The baselines are also not the SOTA techniques as for example the QM9 molecule regression results or semi-supervised graph clustering techniques. An advantage of normalizing flows is to reduce the memory footprints of GNNs, but this is not illustrated in any large graph problems. Also, what is the speed/complexity of GNFs compared to GNNs?

Reviewer 4



The paper is novel on bridging fields between normalizing flow and graph neural networks, but all the sub-parts are not novel. This work provides diverse experiments. However, every task is very simple and no significant performance boost on any of them. The improvements on the supervised tasks are very marginal compared to vanilla GNN. Over simplified dataset on Density estimation which is an important application. No experiments on other density estimation benchmarks. Graph Auto-Encoders part is not core contributions, but some engineering implementation.

[Author Response · NeurIPS 2019]

We thank the reviewers for their detailed comments. We are glad to see a generally positive assessment of our work. The main aim of our work is to develop reversible graph neural network models, called Graph Normalizing Flows (GNFs) which can be used for both supervised learning and unsupervised learning. On supervised tasks, we show that the GNF model exhibits performance comparable to a regular GNN model while providing the advantage of a significantly lower memory footprint. In the unsupervised setting, we develop a permutation-invariant generative model for generating entire graphs in parallel, and also demonstrate the applicability of the model for more accurate density estimation.

We agree with the reviewers that scaling the model to larger graphs is an important problem and are actively working in this area. We will report larger-scale results in the final draft. As the reviewers highlight, the model is a novel approach to this challenging problem and possesses many interesting and useful properties. We believe it will be of great interest to the NeurIPS community. Below, we address specific reviewer comments.

**R1, R3: Memory footprint** We first provide a more rigorous theoretical derivation for the memory footprint and then provide some quantitative results. Let us assume that the node feature dimension is $d$, and the maximum number of nodes in a graph is $N$. Let us assume weights (parameters) of the message passing function is a matrix of size $W$. For simplicity, assume a parameter-free aggregation function that sums over messages from neighbouring nodes. Finally, assume that the final classifier weights are $C$ in size. Suppose we run $K$ message passing steps. Total memory that needs to be allocated for a run of GNN (ignoring gradients for now; gradients will scale by a similar factor) is $W + C + K \times N \times d$ (= memory allotted to weights + intermediate graph-sized tensors generated + adjacency matrix). For a GNF, the total memory is $W + C + N \times d$. Note the *lack* of multiplicative dependence on the number of message passing steps in the latter term.

As a quantitative example, consider a semi-supervised classification task on the Pubmed network ($N = 19717, d = 500$). We assume that the message passing function for a GNN is as follows: $\text{FC}(500) \rightarrow \text{ReLU}() \rightarrow \text{FC}(750) \rightarrow \text{ReLU}() \rightarrow \text{FC}(500)$. Each of the functions $F_1(\cdot)$ and $F_2(\cdot)$ (please see Figure 1 in the paper for notation) in the corresponding GNF have the following architecture: $\text{FC}(250) \rightarrow \text{FC}(750) \rightarrow \text{FC}(250)$. We can compute the total memory allocated to weights/parameters: $W_{GNN} = 500 \times 750 + 750 \times 500, W_{GNF} = 2 \times (250 \times 750 + 750 \times 250)$. We perform $K = 5$ message passing steps for Pubmed. So, the amount of memory allocated to intermediate tensors in a GNN is $19717 \times 500 \times 5 + 19717 \times 750 \times 5$, and correspondingly for a GNF is $19717 \times 500$. Summing up, the overall memory requirements are: GNN = **945.9 M** and GNF = **80.2 M**. Hence, in this case, GNFs are at least $\geq \mathbf{10}\times$ memory efficient than GNNs. Further, we use self-attention in our experiments, which scales according to $\mathcal{O}(N^2)$. GNNs will store attention affinity matrices for each message passing step. In this case, a similar argument can show that this causes a difference of **11G** memory. When using 12G GPU machines, this difference is significant. We will add in a table of memory consumption on all the datasets reported in the paper in the final version.

**R3, R5: results are marginally better than the baselines (QM9). The baselines are also not the SOTA techniques; no significant performance boost on any of them; The improvements on the supervised tasks are very marginal compared to vanilla GNN.** The main aim of the GNF model in the supervised setting is to reduce the memory consumption significantly while providing comparable performance to their GNN counterparts. We agree that the QM9 results are not SOTA, however, we made sure that the comparison performed is fair – the GNN and the GNF architectures were identical with only reversibility being the exception. We will add a comparison between GNF and SOTA GNN techniques (like Graph Attention Networks and any other that the reviewer suggests) in the final version.

**R2: Have you considered balancing the loss for positive and negative edges?** This is a good suggestion. We did investigate the error distribution (false positives and false negatives) and found it to be roughly evenly distributed. In cases where the imbalance does hurt though, this idea could certainly be helpful.

**R3: Also, what is the speed/complexity of GNFs compared to GNNs?** GNFs require 1 extra forward pass during backpropagation as the forward tensors need to be computed for gradient propagation. So, the overall run of a GNF consists of 2 forward passes for each backward pass as compared to 1 forward pass and 1 backward pass for a GNN. We note however, this (relatively cheap) 1 additional forward pass comes at a huge memory benefit.

**R5: Over simplified dataset on Density estimation which is an important application. No experiments on other density estimation benchmarks.** In section 5.1, we demonstrate the effectiveness of GNFs for *structured density estimation*. The goal of structured density estimation is to model densities of a set of inputs. In our experiment (Figure 2), we model densities of a set of four points – one drawn from each Gaussian. Figure 2 shows that modeling densities of points together can outperform modeling per-example (e.g., iid) densities, using a RealNVP, independently. We are not aware of any standard benchmark for this task however if the reviewer has any suggestions, then we can certainly perform experiments on more complicated datasets.

[Meta-Review · NeurIPS 2019]

The reviewers agree that this paper has some interesting ideas, and makes a useful addition to the literature. The reviewers had a few concerns though, and please ensure that the changes suggested in the author response are included in the camera-ready version.